# Nonlinear Relationship between Temporomandibular Joint Disc Displacement Distance and Disc Length: A Magnetic Resonance Imaging Analysis

**DOI:** 10.3390/jcm11237160

**Published:** 2022-12-01

**Authors:** Qinlanhui Zhang, Zheng Ye, Yange Wu, Yufan Zhu, Jiaqi Liu, Wenke Yang, Chengxinyue Ye, Sophie Lau Rui Han, Jun Wang, Xin Xiong

**Affiliations:** 1State Key Laboratory of Oral Diseases and National Clinical Research Center for Oral Diseases, Department of Orthodontics, West China Hospital of Stomatology, Sichuan University, Chengdu 610041, China; 2State Key Laboratory of Oral Diseases and National Clinical Research Center for Oral Diseases, Department of Temporomandibular Joint, West China Hospital of Stomatology, Sichuan University, Chengdu 610041, China; 3Department of Radiology, West China Hospital, Sichuan University, Chengdu 610041, China

**Keywords:** temporomandibular joint, anterior disc displacement, magnetic resonance imaging, disc morphology

## Abstract

Objective: to explore the association between the distance of disc displacement and disc morphology in patients with temporomandibular disorders (TMDs). Methods: a total of 717 joints in 473 subjects were enrolled in this cross-sectional study. The magnetic resonance imaging (MRI) of each patient was evaluated for temporomandibular joint (TMJ) disc morphology classification and position. The distance of the disc displacement and disc length were measured for smoothing spline prediction. A stratified analysis was performed based on the types of disc positions. The disc width and length-width ratio (L/W) were also measured. Descriptive statistics, one-way analysis of variance, smoothing spline analysis, threshold analysis, and two piecewise linear regression were performed to investigate the association between the displacement distance and length of discs. Results: the differences in displacement distance among morphological categories and among different disc positions were statistically significant. Nonlinear relationships were found between distance and length in all subjects. Two turning points of distance (−1.8 mm and 1.7 mm) were found, dividing the curve into three segments. Disc width and L/W were significantly different among discs in the three segments of the curve. The correlation coefficient (β) for the three segments were as follows: −0.6 [95% confidence interval (CI) = −0.9 to −0.3, *p* < 0.001], 0.0 (95% CI = −0.1 to 0.0, *p* = 0.027), and −0.7 (95% CI = −0.8 to −0.7, *p* < 0.001). Nonlinear relationships were also found between the distance and length in cases with anterior disc displacement (ADD), anterior disc displacement with reduction (ADDWR), and without reduction (ADDWoR). Conclusion: the turning points of the disc displacement distance may be considered as a potential reference value for high-risk disc deformation and ADD. Disc length decreases sharply with anterior disc displacement when the disc displacement distance is over 1.7 mm. Prospective and long-term studies are required to clarify the natural course of the disc at different stages of the regression curve.

## 1. Introduction

Temporomandibular disorders (TMDs) are a group of diseases affecting the temporomandibular joint (TMJ), masticatory muscle, and adjacent structures [1]. Anterior disc displacement (ADD) is one of the most common subtypes of TMDs [2], which can occur in all age groups, and is predominantly seen in adult women [3]. Patients with ADD often present with various symptoms, such as pain, joint sounds, and impaired jaw movement [4], which can significantly affect an individual’s quality of life [5]. Moreover, ADD might progress over time [6] and lead to condylar osseous destruction [7]. ADD can be further divided into anterior disc displacement with reduction (ADDWR) and without reduction (ADDWoR). Compared to ADDWR, ADDWoR often presents with severe symptoms and higher difficulty in treatment [8,9]. There is another type of disc displacement called posterior disc displacement (PDD). Although PDD is reported to be very rare [10], symptoms such as pain, clicking, and locking can still occur [11].

Various options of management have been suggested for disc displacement, including conservative treatment [12,13] (such as medication, splint, and physical therapy) and surgical treatment [14] (such as arthrocentesis and disc repositioning by arthroscopy or open surgery). However, to date, a consensus has yet to be reached on a treatment protocol, and the therapeutic outcomes of different methods on varying degrees of disc deformation are still unclear. Identifying the type of disc displacement and its relationship with TMJ disc alteration is crucial for treatment planning. Several studies have shown that the morphological changes of TMJ discs were highly associated with the type of disc displacement [15,16,17]. In our previous study, a corresponding relationship was found between the types of disc morphology and disc position [18], where the greater the degree of articular disc folding, the easier it is to develop nonreducing disc displacement. Furthermore, an increase in disc displacement resulted in more severe TMJ disc deformation [6,19]. However, limited studies quantify the morphological change in TMJ discs and their relationship with the degree of disc displacement. 

Magnetic resonance imaging (MRI) plays an indispensable role in the diagnosis of ADD, which is regarded as the gold standard for TMJ disc examination according to the research diagnostic criteria for temporomandibular disorders (RDC/TMD) [20]. As a radiation-free imaging approach [21], MRI can provide precise anatomical information on TMJ disc morphology and position, including the distance of the disc displacement and disc length, as well as other quantitative measurements (disc width and length-width ratio) of TMJ discs. Therefore, the aims of the present study are first, to determine the corresponding effect of disc displacement distance on the disc length and other morphological variables; second, to investigate the associations among disc displacement distance, disc morphology classification, and the types of disc position; and third, to explore the potentially high-risk disc position for disc deformation.

## 2. Materials and Methods

### 2.1. Study Population

This cross-sectional study was approved by the Institution Review Board of West China Hospital of Stomatology, Sichuan University. The patients who sought for TMDs treatment at the Temporomandibular Joint department of West China Hospital of Stomatology were enrolled between February 2019 and December 2021. MRI images for each patient were acquired on the first visit for the evaluation of the disc status of both TMJs.

The inclusion criteria for patients were as follows: (1) had a diagnosis of TMDs according to the RDC/TMD Axis I [22]; (2) met the age restriction of 18 to 60 years old; (3) permanent dentition with no missing teeth; (4) had no previous TMJ treatment, orthodontic treatment, or craniofacial treatment. The exclusion criteria for patients were as follows: (1) a history of infection and injuries with the TMJs; (2) systematic diseases; (3) severe artifacts on MRI images or unsuitable for quantitative measurements of discs. A total of 473 patients were randomly selected and enrolled in the present study, whose 946 joints were included. Two hundred and twenty-nine joints were excluded due to the poor quality of the MRI images and unsuitability for disc quantitative measurements, resulting in a final total of 717 joints enrolled. The minimum sample size was calculated using GPower software on the condition that α was set to 0.05, (1 − β) was set to 0.8, and the effect size *f* was set to 0.25. The minimum total sample size was calculated as 314, which is smaller than the number of joints included in the study. 

### 2.2. MRI Protocol

All MRI images were acquired using a 1.5-T scanner (Achieva, Philips Healthcare, Amsterdam, The Netherlands) with an 8-channel head coil. The T2-weighted images were obtained in the oblique sagittal plane, which was determined by the long axis of the condyle. The imaging parameters were as follows: repetition time = 2000 ms, echo time = 80 ms, slice thickness = 1 mm, slice gap = 0.3 mm, the field of view = 160 × 160 mm^2^, and matrix = 256 × 256. The teeth position was maintained at a maximum intercuspal position in the closed mouth position, and the maximum voluntary mouth opening of each patient was required in the open mouth position during MRI scanning. 

### 2.3. Image Analysis

All image interpretation and measurement were achieved using RadiAnt DICOM Viewer (Medixant, Poznań, Poland). The sagittal slice with the most severe deformation and displacement of the TMJ discs in the closed mouth position was chosen for evaluation. For all quantitative variables and categorical variables, each MRI image was interpreted by an experienced TMJ specialist. Fifty MRI images were then randomly selected from the enrolled joints and remeasured by the same examiner at least one month apart.

#### 2.3.1. Determination of Disc Displacement

The evaluation of disc displacement was made in the closed-mouth position. ADD was defined as the most posterior edge of the disc located anteriorly to the 12:00 clock position relative to the long axis of the condyle. In contrast, PDD was defined as the most posterior edge of the disc located posteriorly to the 12:30 clock position relative to the long axis of the condyle. The disc was otherwise determined as being in a normal position. ADD with or without reduction was determined in the maximum opening mouth position. When the intermediate zone of the disc was located between the condylar articular surface and the articular eminence, the cases were classified as ADDWR. Conversely, the cases were classified as ADDWoR [23,24]. The kappa coefficient was calculated for intra-examiner reliability, and the kappa value was 0.85 [95% confidence interval (CI), 0.78 to 0.92].

#### 2.3.2. Quantitative Measurements of Disc Morphology

The long axis of the condyle was determined using the two-step method described by Nebbe et al. and Xie et al. [25,26]. The marker points of the disc and the condyle were first identified (Figure 1). Point A was determined as the most anterior point of the disc, and point C was determined as the most posterior point of the disc. Point B was the midpoint of the intermediate zone of the disc. Point D was the intersection point between the long axis of the condyle and the contour of the condylar bone cortex. Lastly, the disc displacement distance and disc length, as well as other linear variables, were measured and calculated. The length of the disc was defined as the sum of segment AB and BC, while the distance of the disc displacement was defined as the length of the CD. The distance of the disc displacement had a positive value for cases with ADD and a negative value for cases with a normal disc position and PDD. The width of the disc was measured at point B. Length-width ratio (L/W) was then calculated as the length divided by the width. The intra-examiner intraclass correlation coefficient (ICC) of all the measurements was calculated. All intra-examiner ICCs were over 0.80.

#### 2.3.3. Classification of Disc Morphology

Disc morphology was classified into three categories as described by our previous study [18]. Specifically, Class 1 discs referred to a rather normal morphology, Class 2 discs showed a mild disc deformation, and Class 3 discs were severely damaged or exhibited a deformed disc morphology (Figure 2). The kappa coefficient was calculated for intra-examiner reliability, and the kappa value was 0.90 (95% CI, 0.85 to 0.94).

### 2.4. Statistical Analysis

All analyses were performed using R (http://www.R-project.org, accessed on 1 June 2022), Empower software (www.empowerstats.com, accessed on 1 June 2022, X&Y Solutions, Inc., Boston, MA, USA) and SPSS (version 21.0, SPSS Institute Inc., Chicago, IL, USA). The mean and standard deviation were calculated for the descriptive statistics. A one-way analysis of variance (ANOVA) was used to investigate the differences in the disc morphological measurements and displacement distance among different disc position groups and different disc morphology categories. Spearman’s rank correlation coefficient was used to investigate the correlation of disc quantitative measurements with the disc position. Smoothing spline analysis was performed to estimate the independent relationship between the disc displacement distance and other disc lengths with an adjustment for age and sex. Additionally, threshold analysis and a two-piecewise linear regression model were used to examine the turning points and the threshold effect of the disc displacement distance on disc length. Stratified analysis based on the disc position was performed to explore the independent relationship in sub-groups. *p* < 0.05 was considered statistically significant.

## 3. Results

### 3.1. Demographic Characteristics

A total of 473 patients (398 females and 75 males) with a mean age of 29.28 ± 11.35 years were included in the present study (Table 1). Discs with ADDWoR accounted for the largest proportion (328 discs, 45.75%), while 218 (30.40%) discs were identified as ADDWR, 143 discs (19.94%) were in normal position, and 28 discs (3.91%) were PDD. For the classification of disc morphology, 204 discs (28.45%) were Class 1, 280 discs (39.05%) were Class 2, and 233 discs (32.50%) were Class 3. As shown in Table 2, significant differences in age were found among different disc morphology categories (*p* < 0.001, Class 1 > Class 2 > Class 3), as well as different disc position groups (*p* < 0.001, normal position > ADDWR > ADDWoR).

### 3.2. MRI Evaluations

The distance of the disc displacement, disc length, disc width, and L/W exhibited significant differences among the three-disc morphological categories (Table 3). Similarly, quantitative measurements of the disc exhibited significant differences among the normal disc position, ADDWR, ADDWoR, and PDD (Table 4). After adjustment for age and sex using a generalized additive model, the differences in disc quantitative measurements among the groups were statistically significant (Table 5).

### 3.3. Association of Disc Displacement and Morphology

A moderate negative correlation was found between the displacement distance and disc length (r = −0.549, *p* < 0.001). The L/W ratio also showed a moderate negative correlation with displacement distance (r = −0.501, *p* < 0.001). The disc width showed a weak positive correlation with disc displacement distance (r = 0.250, *p* < 0.001).

Spline smoothing plots suggested a nonlinear relationship between the distance of the disc displacement and disc length. Two turning points were found in the curve, which were at −1.8 mm and 1.7 mm, respectively. When the distance was less than −1.8 mm, the disc length decreased sharply with a correlation coefficient (β) of −0.6 (95%CI = −0.9 to −0.3, *p* < 0.001) as the distance increased. The β-value suggested that the disc length decreased by 0.6 mm for every 1 mm increase in the displacement distance. Between the two turning points, an approximately horizontal line was shown. When the distance increased up to 1.7 mm, the disc length decreased sharply again (β = −0.7, 95%CI = −0.8 to −0.7, *p* < 0.001) (Figure 3). The disc width (*p* < 0.001) and L/W (*p* < 0.001) showed significant differences among the three segments of the curve. 

Smoothing spline analysis was then used to investigate all subjects with ADD. A nonlinear relationship was also found between the distance of the disc displacement and disc length in ADD cases. The turning points were 1.4 mm and 7.5 mm, respectively. When the distance was less than 1.4 mm, the disc length slightly increased as the disc moved forward (β = 0.4, 95%CI = −0.1 to 0.8, *p* < 0.001). Between the two turning points, a steep decrease in the disc length was found as the distance increased (β = −0.8, 95%CI = −0.9 to −0.8, *p* < 0.001). A horizontal line was shown when the distance was larger than 7.5 mm (Figure 4). The disc width (*p* < 0.001) and L/W (*p* < 0.001) showed significant differences among the three segments of the curve.

In addition, a stratified analysis was performed among ADD cases. In cases with ADDWR, a nonlinear relationship was found between the disc displacement distance and disc length (Figure 5). Before the turning point at 2.0 mm, the length remained almost constant as the distance became larger. After the turning point, the length decreased steeply as the distance enlarged (β = −0.7, 95%CI = −1.0 to −0.4, *p* < 0.001) (Figure 5). In cases with ADDWoR, a nonlinear relationship between the distance and length and a turning point at 7.1 mm was found. When the distance was less than 7.1 mm, the disc length was rapidly reduced as the distance enlarged (β = −0.9, 95%CI = −1.0 to −0.8, *p* < 0.001). When the disc was displaced beyond 7.1 mm, the trend of the curve was relatively flat (Figure 6).

The threshold effect of the disc displacement distance on disc length was shown in Table 6.

## 4. Discussion

Based on MRI, the present study investigated the quantitative and qualitative variables of TMJ discs, including the disc length, displacement distance, position, and morphology classification. Disc width and L/W were also measured. 

Similar to past studies [27,28], the female subjects dominated the present study as they are more predisposed toward TMJ internal degeneration induced by female sex hormones [29,30] and gender-related differences in joint structure [31]. Patients with ADDWoR and patients with Class 3 discs were significantly older. This might be due to age-related changes in the TMJ discs. Disc deformation is more frequently seen in the elderly, which is considered a common pathophysiological change [32].

In the present study, when the positions of the discs were normal, Class 1 morphology was mainly exhibited, whereas ADDWoR discs were mainly Class 3, which is consistent with our previous study [18]. Statistically significant differences were found in the distance of the disc displacement among the three-disc morphological categories. Compared to ADDWR discs, ADDWoR discs were displaced further forward. The results suggested that TMJ discs showed more severe deformation and a higher tendency of ADDWoR with the increase in the displacement distance, which was in accordance with previous studies [6,19]. The cases with tooth loss were excluded from the study. Tooth loss, especially in the posterior segment of the dental arch, is apt to cause mandibular rotation and changes to the condyle position, which might affect the morphology and position of the TMJ disc. Due to the large age span of the samples included in this study, adjustments for age and sex were performed to examine whether the differences in disc displacement distance were influenced by sex and age. The association of the displacement distance with morphological category and disc position was still statistically significant.

Although several studies have reported the morphological changes of discs in ADDWR and ADDWoR [16,33,34,35], none of them have investigated the association between displacement distance and disc morphological variables. The quantitative measurements, including disc length, width, and L/W, showed a significant association with the displacement distance. With an increase in the displacement distance, the discs exhibited smaller lengths, larger widths, and smaller L/W. The results indicated that anterior displacement and morphological deformation became severe synchronously when the disc was pushed further forward.

In the present study, the disc length shortened as the displacement distance increased. There were two turning points dividing the curve of the distance-length into three segments (Figure 3). Before the distance of −1.8 mm and after the distance of 1.7 mm, the effect of the distance change on the disc length was more significant. Due to the association of a large displacement distance with a nonreducing disc [19], the turning point might be valuable as a reference value for the disc morphology to estimate a possible disc position. Peroz et al. [36] reported an average disc length of 12.8 mm in asymptomatic volunteers. The length corresponding to the turning points on the curve was approximately 11 mm in the present study, which is smaller than the average disc length in asymptomatic volunteers. Compared to the discs in a normal position, the discs in the further forward position were considerably compressed during the condylar movement, which destroyed the structure and impaired the function of the discs [37,38], resulting in shrunken disc morphology. This effect of compression-only became destructive when the distance was larger than 1.7 mm, forming a steep descent on the curve. The section of the curve with a distance smaller than −1.8 mm represented the posterior edge of the disc, which is located in a position posterior to the condylar long axis. The discs in this section of the curve were determined as PDD or in the normal position. The farther the posterior edge of the disc is located, the larger the disc length. Therefore, this section of the curve might be formed because of the change in the disc length but not the change in disc position. Since PDD cases were concentrated on the left side of the first turning point of the curve, the steepness of the front section of the curve could be mainly caused by PDD discs.

In the analysis focused on subjects with ADD, disc length also showed nonlinear relationships with disc displacement distance. The disc length decreased more steeply when the displacement distance was between two turning points (Figure 4). The first turning point divided all ADD discs into a group of long and narrow discs where ADDWR and Class 1 or 2 were more commonly seen and a group of short and thick discs where ADDWoR and Class 3 were more commonly seen. This result was consistent with previous studies [18,19]. Thus, for cases with ADD, the distance of the disc displacement could be used as a predictor for the degree of disc deformation and the capability of reduction. Previous studies reported that ADD discs become displaced further forward over time, especially in nonreducing cases [6,39]. Although the importance of the disc position and function remains controversial [40], and there is no general agreement on the treatment protocol for ADD, the deformed and displaced discs should be treated as soon as possible to prevent the aggravation of disc displacement and the long-term detrimental effect of ADD on the condyle osseous status and mandibular growth [7,41,42], especially for cases after the first turning point of disc displacement. However, after the second turning point, the disc length only changed slightly as the displacement distance changed. This suggested that the disc had been deformed to the largest limit. Subsequently, the discs could only move forward with the compression of the condylar, and the shape could no longer undergo an obvious change. Although only the middle segment of the curve had an adjusted *β* value which was statistically significant, the trend of the curve still suggested that the aggravation of the disc anterior displacement imposed a high risk of disc deformation, which should be paid attention to in clinical diagnosis and treatment.

The chief complaints of most patients in the study are pain, clicking, locking, etc., which are common symptoms of ADD. Although ADDWR can remain stable for years [43] due to the adaptive processes of retrodiscal fibrosis [44], it is still possible to develop into ADDWoR [6]. Greater disc displacement is associated with more severe symptoms and a lower possibility of disc reduction, which leads to changes in treatment methods. In ADD cases (Figure 4), the segment before point A represents a mild anterior displacement, which may not need management if no symptom is shown [43]. In the segment between points A and B, which is mostly composed of ADDWR cases and acute ADDWoR cases, conservation approaches such as the anterior repositioning splint [45] could be used for relieving symptoms and preventing progress [46], even repairing and regenerating the condylar bone [47]. This is due to the fact that cases in this segment of the curve may be still salvageable, and it is possible for conservation approaches to achieve the goal of disc reduction. However, for severe ADDWoR cases with extremely serious damage on the discs, which is concentrated in the segment after point B, different treatment goals should be set, and different treatment methods should be adopted to prevent the occurrence of more serious complications such as condylar bone destruction [48]. Invasive methods may be considered if the attempts of conservative treatments fail [49].

A nonlinear relationship was found between the displacement distance and disc length in ADDWR cases. There was an approximately horizontal line when the disc displacement distance was smaller than 2.0 mm, and the disc length changed sharply when the distance was larger than 2.0 mm (Figure 5). However, most of the ADDWR cases were concentrated in the horizontal section on the curve, which indicated that the length change in the disc was not significant in most ADDWR cases. In cases with ADDWoR, the relationship was still nonlinear, with a relatively flat slope after the displacement distance increased up to 7.1 mm (Figure 6). This could be attributed to the increase in disc compression during condylar movement, which corresponds to the pressure exerted by the condyle, leading to a gradual reduction in the disc length in cases with ADDWR. The disc position remains unchanged until it is compressed beyond its capacity, resulting in substantial anterior displacement, and disc displacement shifts from reducing to nonreducing. However, the shrinking of the disc morphology is finite. When the discs were shortened and deformed to a certain extent, the effect of displacement distance on disc morphological changes was invalid.

There were some limitations in the present study. Firstly, the cross-sectional study failed to provide cause-effect information on disc distortion and displacement; thus, future prospective studies are needed in this regard. Secondly, a long-term investigation is required to estimate the possible outcomes in the natural course of the discs on both sides of the turning points. Thirdly, since the sample size decreased after stratified analysis based on the disc position, which resulted in a large dispersion of some segments of the curve, the sample size should be expanded in future research to confirm whether the relationship is linear or nonlinear. Fourthly, MRI images of some of the joints were excluded from the present study due to poor quality and unsuitability for disc quantitative measurements, which may cause a loss of information and subsequent bias. Furthermore, cases with tooth loss are excluded from the present study. Therefore, the conclusions of this study are limited to patients without missing teeth. It is necessary to conduct further studies, including patients with tooth loss, to broaden applicability. Finally, all subjects in the study were only enrolled patients seeking TMJ treatment. Further studies that recruit samples from the general population are needed to broaden the applicability of the correlation between disc morphology and position. At the same time, other factors, such as malocclusion and craniofacial features, should also be considered and reported to give a full picture of the study sample.

## 5. Conclusions

The distance of the disc displacement is greater in severely deformed TMJ discs and ADDWoR discs. As the discs move farther forward, the shortening and thickening of the discs become more significant. A non-linear relationship was found between the disc length and distance of disc displacement. The turning points of the displacement distance were −1.8 mm and 1.7 mm, dividing the curve into three sections (a steep slope at first, followed by a flat curve, and a steep slope again), which indicated the need to focus on the critical stage of this specific disc morphology. Future prospective and long-term studies are required to better understand the possible outcomes of the discs at different stages of the curves.

## Figures and Tables

**Figure 1 jcm-11-07160-f001:**
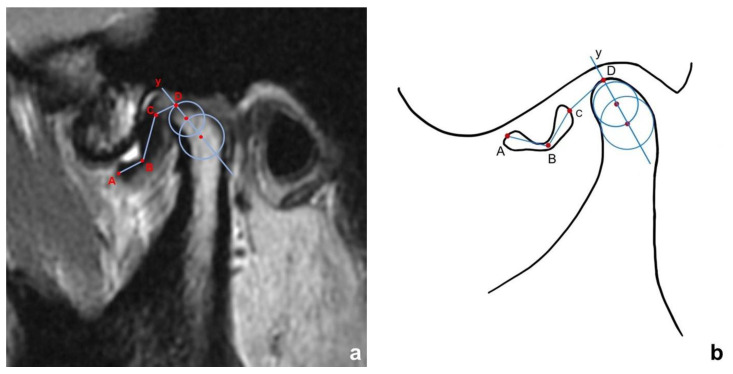
Measurement of quantitative disc variables. (**a**) true measurement on MRI; (**b**) measurement in diagrammatic sketch. Two circles were drawn to the internally tangent of the condyle. y, the long axis of the condyle; A, the midpoint of the most anterior edge of the disc; B, the midpoint of the intermediate zone; C, the midpoint of the posterior edge of the disc; D, the intersection of condylar long axis and contour of condylar cortical bone. Disc length equals to AB + BC; disc width is measured at point B; length-width ratio (L/W) is calculated as length divided by width; distance between the disc and condylar head equals CD, which equals the disc displacement distance in ADD cases.

**Figure 2 jcm-11-07160-f002:**
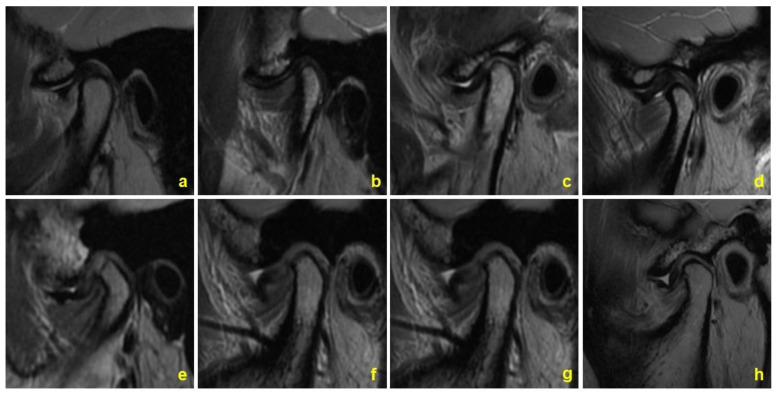
Huaxi classification for disc morphology. (**a**) Class 1 disc, biconcave disc with a L/W larger than 10; (**b**) Class 1 disc, biplanar disc with a L/W larger than 10; (**c**) Class 2 disc, disc bending upwards with a bending angle larger than 120°; (**d**) Class 2 disc, disc with no bending and a L/W smaller than 10; (**e**) Class 3 disc, disc bending twice; (**f**) Class 3 disc, disc bending downwards, (**g**) Class 3 disc, disc bending upwards with a bending angle smaller than 120°; (**h**) Class 3 disc, folded disc.

**Figure 3 jcm-11-07160-f003:**
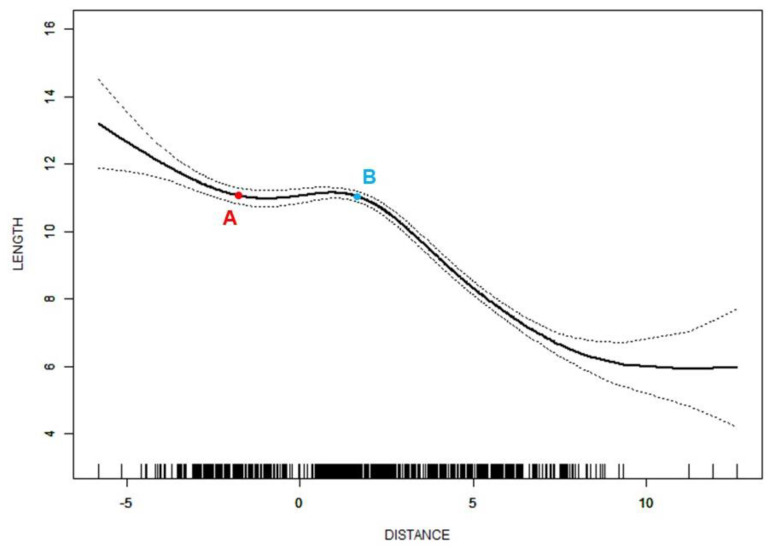
Nonlinear relationship between distance of disc displacement (mm) and disc length (mm). Adjusted for age and sex; A—the turning point of −1.8 mm; B—the turning point of 1.7 mm.

**Figure 4 jcm-11-07160-f004:**
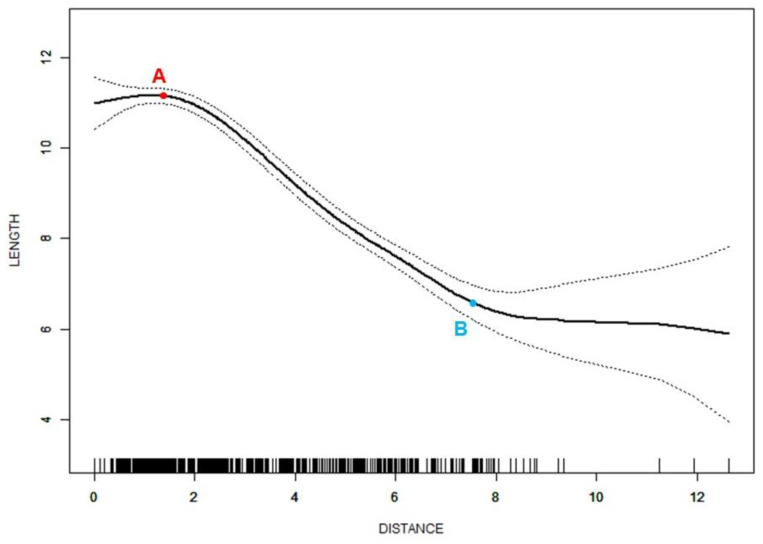
Nonlinear relationship between distance of disc displacement (mm) and disc length (mm) in cases with ADD. Adjusted for age and sex; ADD—anterior disc displacement; A—the turning point of 1.4 mm; B—the turning point of 7.5 mm.

**Figure 5 jcm-11-07160-f005:**
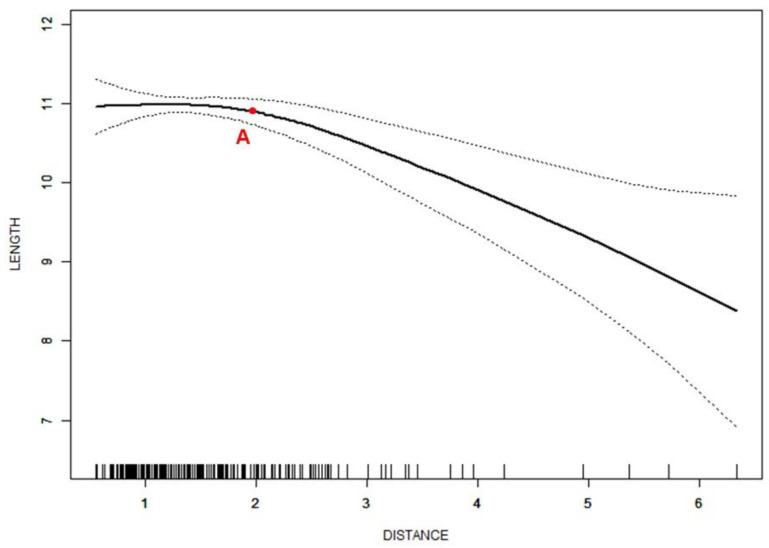
Nonlinear relationship between distance of disc displacement (mm) and disc length (mm) in cases with ADDWR. Adjusted for age and sex; ADDWR—anterior disc displacement with reduction; A—the turning point of 2.0 mm.

**Figure 6 jcm-11-07160-f006:**
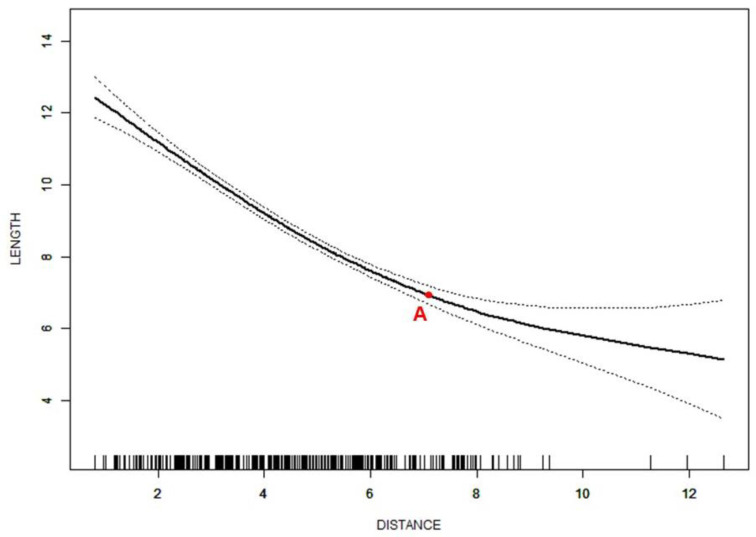
Nonlinear relationship between distance of disc displacement (mm) and disc length (mm) in cases with ADDWoR. Adjusted for age and sex; ADDWoR—anterior disc displacement without reduction; A—the turning point of 7.1 mm.

**Table 1 jcm-11-07160-t001:** Details of the study population.

	Normal Position (*n*= 143)	ADDWR (*n* = 218)	ADDWoR (*n* = 328)	PDD (*n* = 28)
Gender				
Female (*n*, %)	115 (80.42)	188 (86.24)	282 (85.98)	19 (67.86)
Male (*n*, %)	28 (19.58)	30 (13.76)	46 (14.02)	9 (32.14)
Morphology classification				
Class 1 (*n*, %)	109 (76.22)	67 (30.73)	6 (1.83)	22 (78.57)
Class 2 (*n*, %)	34 (23.78)	144 (66.06)	96 (29.27)	6 (21.43)
Class 3 (*n*, %)	0 (0.0)	7 (3.21)	226 (68.90)	0 (0.0)

ADDWR—anterior disc displacement with reduction; ADDWoR—anterior disc displacement without reduction; PDD—posterior disc displacement.

**Table 2 jcm-11-07160-t002:** Age data of study participants.

	Normal (*n* = 143)	ADDWR (*n* = 218)	ADDWoR (*n* = 328)	PDD (*n* = 28)	Class 1 (*n* = 204)	Class 2 (*n* = 280)	Class 3 (*n* = 233)
Mean ± SD	26.36 ± 10.18	28.96 ± 10.20	33.85 ± 12.37	26.28 ± 11.21	26.38 ± 10.60	28.28 ± 9.75	32.78 ± 12.45
Median	24	27	31	23	24	26	30
Interquartile range	19, 30	23, 32	23, 45	19, 30	19, 30	22, 32	23, 43

SD—standard deviation; ADDWR—anterior disc displacement with reduction; ADDWoR—anterior disc displacement without reduction; PDD—posterior disc displacement.

**Table 3 jcm-11-07160-t003:** Differences in disc quantitative measurements among the three-disc morphology categories.

	Class 1 (*n* = 204)	Class 2 (*n* = 280)	Class 3 (*n* = 233)	*p*-Value	Multiple Comparison
Distance	−1.04 ± 2.18	1.71 ± 2.22	5.13 ± 2.00	<0.001	1 < 2 < 3
Length	11.13 ± 1.27	10.65 ± 1.55	8.28 ± 2.18	<0.001	3 < 2 < 1
Width	0.89 ± 0.18	1.29 ± 0.37	1.56 ± 0.65	<0.001	1 < 2 < 3
L/W	12.96 ± 2.78	8.14 ± 2.20	6.23 ± 2.99	<0.001	3 < 2 < 1

Distance—distance of disc displacement; L/W—length-width ratio.

**Table 4 jcm-11-07160-t004:** Differences in disc quantitative measurements among normal disc position, ADDWR, ADDWoR, and PDD.

	Normal (*n* = 143)	ADDWR (*n* = 218)	ADDWoR (*n* = 328)	PDD (N = 28)	*p*-Value	Multiple Comparison
Distance	−1.67 ± 0.88	1.63 ± 0.89	4.67 ± 2.03	−3.79 ± 0.66	<0.001	PDD < Normal < ADDWR < ADDWoR
Length	11.05 ± 1.34	10.87 ± 1.27	8.83 ± 2.22	12.05 ± 1.35	<0.001	ADDWoR < ADDWR < PDD Normal < PDD
Width	1.01 ± 0.31	1.19 ± 0.37	1.53 ± 0.60	1.07 ± 0.58	<0.001	Normal < ADDWR < ADDWoR PDD < ADDWoR
L/W	11.85 ± 3.43	9.94 ± 3.03	6.63 ± 2.92	12.03 ± 0.14	<0.001	ADDWoR < ADDWR < Normal ADDWoR < ADDWR < PDD

Distance—distance of disc displacement; L/W—length-width ratio; ADDWR—anterior disc displacement with reduction; ADDWoR—anterior disc displacement without reduction; PDD—posterior disc displacement.

**Table 5 jcm-11-07160-t005:** Differences in disc quantitative measurements after adjusting for age and sex. (**a**) Differences in disc quantitative measurements among the three disc morphology categories after adjusting for age and sex ^a^. (**b**) Differences in disc quantitative measurements among normal disc position, ADDWR, ADDWoR, and PDD after adjusting for age and sex ^a^.

(**a**)					
	**Class 1 (** ***n* = 204)**		**Class 2 (** ***n* = 280)**	**Class 3 (** ***n* = 233)**	***p*-Value**
Distance	−0.96 (−1.26, 10.67)		1.68 (1.43,1.93)	5.09 (4.82, 5.37)	<0.001
Length	11.14 (10.91, 11.38)		10.67 (10.47, 10.87)	8.26 (8.04, 8.48)	<0.001
Width	0.87 (0.81, 0.93)		1.39 (1.34, 1.44)	1.58 (1.52, 1.63)	<0.001
L/W	13.04 (12.67, 13.40)		8.14 (7.83, 8.45)	6.15 (5.18, 6.49)	<0.001
(**b**)					
	**Normal (** ***n* = 143)**	**ADDWR (** ***n* = 218)**	**ADDWoR (** ***n* = 328)**	**PDD (** ***n* = 28)**	***p*-Value**
Distance	−1.70 (−1.96, −1.45)	1.63 (1.43, 1.84)	4.68 (4.51, 4.85)	−3.82 (−4.38, −3.25)	<0.001
Length	11.10 (10.81, 11,40)	10.87 (10.64, 11.11)	8.81 (8.61, 9.00)	12.02 (11.36, 12.68)	<0.001
Width	0.98 (0.90, 1.06)	1.19 (1.13, 1.25)	1.54 (1.49, 1.59)	1.05 (0.88, 1.23)	<0.001
L/W	11.94 (11.43, 12.45)	9.96 (9.55, 10.37)	6.58 (6.25, 6.92)	12.01 (10.87, 13.15)	<0.001

^a^ Data in the table: adjust mean (95% confidence interval), only measurements with statistical differences are illustrated; Distance—distance of disc displacement; L/W—length-width ratio; ADDWR—anterior disc displacement with reduction; ADDWoR—anterior disc displacement without reduction; PDD—posterior disc displacement.

**Table 6 jcm-11-07160-t006:** Threshold effect analysis of disc displacement distance on disc length using two piece-wise linear regression.

	Adjusted * *β* (95% CI)	*p*-Value	Difference ^†^ in Adjusted * *β* (95% CI)	*p*-Value
All subjects				
Distance ≤ −1.8 mm	−0.6 (−0.9, −0.3)	<0.001	Effect 2-1: 0.6 (0.4, 0.8) Effect 3-2: −0.7 (−0.8, −0.6)	<0.001 <0.001
−1.8 mm < Distance ≤ 1.7 mm	0.0 (−0.1, 0.0)	0.027		
1.7 mm < Distance	−0.7 (−0.8, −0.7)	<0.001		
ADD subjects				
Distance ≤ 1.4 mm	0.4 (−0.1, 0.9)	0.166	Effect 2-1: −1.2 (−1.7, −0.7) Effect 3-2: 0.8 (0.4, 1.2)	<0.001 <0.001
1.4 mm < Distance ≤ 7.5 mm	−0.8 (−0.9, −0.8)	<0.001		
7.5 mm < Distance	−0.0 (−0.4, 0.3)	0.939		
ADDWR subjects				
Distance ≤ 2.0 mm	0.1 (−0.3, 0.5)	0.512	Effect 2-1: −0.8 (−1.4, −0.2)	0.009
2.0 mm < Distance	−0.7 (−1.0, −0.4)	<0.001		
ADDWoR subjects				
Distance ≤ 7.1 mm	−0.9 (−1.0, −0.8)	<0.001	Effect 2-1: 0.8 (0.4, 1.2)	<0.001
7.1 mm < Distance	−0.1 (−0.4, 0.2)	0.538		

*, adjusted for age and sex; ^†^—difference in effect of distance on length between adjacent sections on the curve; Distance—distance of disc displacement.

## Data Availability

The data presented in this study are available on request from the corresponding author. The data are not publicly available due to ethical reasons.

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
