# Peer review of "Nonlinear Relationship between Temporomandibular Joint Disc Displacement Distance and Disc Length: A Magnetic Resonance Imaging Analysis"

_jcm, 2022, doi:10.3390/jcm11237160_

Round 1

Reviewer 1 Report

This article focuses on the morphological analysis on MRI of the disc appearance according to the position of the mandibular condyle.

This article is clear and well written. However, several points need clarification and discussion:

- an inclusion criteria is to have "permanent dentition with no missing teeth". Please discuss this major inclusion biais within discussion section

- the authors have computed the number of samples with alpha and beta risks, but the other statistical parameters were not given. On which parameter computation was made, standard deviation ... ?

- Why a total of 473 has be obtained while a final total of 717 joints enrolled?

- Why T2 MRI ? Spin echo ? Why not T1 weighted ?

- The following sentence is useless, please delete "To our knowledge, it is the first time that smoothing spline analysis was used to estimate relationships between distance of disc displacement and disc length, where nonlinear relationships were found".

- I would not be as confident as the authors in describing a non-linear relationship in figure (4), 5 and 6 given the high standard deviation on the analysis.

- I would have added on Fig 1 a true mesure on a MRI

Reviewer 2 Report

This manuscript, titled " Nonlinear relationship between temporomandibular joint disc displacement distance and disc length: A magnetic resonance imaging analysis " describes an original clinical study. The assumptions of the research are sound, and the scientific methods and materials are merits. The results provide evidence to prove the hypothesis. But there are poins in the manuscript that need clarification.

In study population selection: this study included patients with a wide age range (18-60 years), and one of the inclusion criteria was " permanent dentition with no missing teeth "; this means that a minority of patients with TMDs in the 50-60 age group who did not lose teeth, met this criterion, and most patients with TMDs in this age group were excluded from this study. The authors need to explain why this criterion was chosen.

There is a lack of information on occlusion in the TMDs patients included in this study, but malocclusion is one of the known factors in the development of TMDs.

In this study, the authors should point out the main symptoms of the clinical diagnosis and discuss more about the significance of changes in MRI morphological features in clinical treatment needs.

Round 2

Reviewer 1 Report

I thank the authors for having perfectly taken into account all the remarks